# Mechanisms Leading to Increased Insulin-Stimulated Cerebral Glucose Uptake in Obesity and Insulin Resistance: A High-Fat Diet and Exercise Training Intervention PET Study with Rats (CROSRAT)

**DOI:** 10.3390/jfmk9020058

**Published:** 2024-03-25

**Authors:** Anna Jalo, Jatta S. Helin, Jaakko Hentilä, Tuuli A. Nissinen, Sanna M. Honkala, Marja A. Heiskanen, Eliisa Löyttyniemi, Tarja Malm, Jarna C. Hannukainen

**Affiliations:** 1MediCity Research Laboratory, University of Turku, Tykistökatu 6 A, FI-20520 Turku, Finland; 2Preclinical Imaging Laboratory, Turku PET Centre, University of Turku, Tykistökatu 6 A, FI-20520 Turku, Finland; 3Doctoral Programme in Clinical Research, University of Turku, FI-20520 Turku, Finland; 4Turku PET Centre, University of Turku, P.O. Box 52, FI-20521 Turku, Finland; 5Research Centre of Applied and Preventive Cardiovascular Medicine, University of Turku, Kiinamyllynkatu 10, FI-20520 Turku, Finland; 6Centre for Population Health Research, University of Turku and Turku University Hospital, Kiinamyllynkatu 10, FI-20520 Turku, Finland; 7Department of Biostatistics, University of Turku and Turku University Hospital, Kiinamyllynkatu 10, FI-20520 Turku, Finland; 8A.I. Virtanen Institute for Molecular Sciences, University of Eastern Finland, Yliopistonranta 8, FI-70210 Kuopio, Finland

**Keywords:** obesity, insulin resistance, glucose uptake, inflammation, hyperinsulinemic euglycemic clamp, molecular imaging

## Abstract

Recent studies have shown that obesity and insulin resistance are associated with increased insulin-stimulated glucose uptake (GU) in the brain. Thus, insulin sensitivity seems to work differently in the brain compared to the peripheral tissues like skeletal muscles, but the underlying mechanisms remain unknown. Regular exercise training improves skeletal muscle and whole-body insulin sensitivity. However, the effect of exercise on glucose metabolism in the brain and internal organs is less well understood. The CROSRAT study aims to investigate the effects of exercise training on brain glucose metabolism and inflammation in a high-fat diet-induced rat model of obesity and insulin resistance. Male Sprague Dawley rats (*n* = 144) are divided into nine study groups that undergo different dietary and/or exercise training interventions lasting 12 to 24 weeks. Insulin-stimulated GU from various tissues and brain inflammation are investigated using [^18^F]FDG-PET/CT and [^11^C]PK11195-PET/CT, respectively. In addition, peripheral tissue, brain, and fecal samples are collected to study the underlying mechanisms. The strength of this study design is that it allows examining the effects of both diet and exercise training on obesity-induced insulin resistance and inflammation. As the pathophysiological changes are studied simultaneously in many tissues and organs at several time points, the study provides insight into when and where these pathophysiological changes occur.

## 1. Introduction

Obesity and sedentary lifestyle are risk factors for the development of insulin resistance and type 2 diabetes (T2D). Insulin resistance is defined as an attenuated response to insulin leading to decreased insulin-stimulated glucose transport and metabolism in adipose tissue and skeletal muscle, as well as impaired suppression of endogenous glucose production in the liver [1]. Recent studies have shown that the brain functions differently compared with skeletal muscle and adipose tissue in obese and/or insulin-resistant humans. Using the 2-deoxy-2-^18^F-fluoro-D-glucose ([^18^F]FDG) positron emission tomography (PET) method, insulin-stimulated brain glucose uptake (GU) has been shown to be increased in obesity and associate negatively with whole-body insulin sensitivity [2,3]. We and others have shown that the increased insulin-stimulated brain GU is reversible and ameliorates after weight loss in morbid obesity and reduces already after short-term exercise training in insulin-resistant subjects [2,4]. These findings have provoked an interest in studying the role of the brain in obesity and T2D. 

Currently, the mechanisms underlying the altered brain glucose metabolism in obesity and insulin resistance are unclear [5]. It has been suggested that the brain can rapidly sense changes in peripheral metabolism and inflammation and that reciprocal interactions exist between the central nervous system and systemic metabolism [6]. The systemic inflammation in obesity and insulin resistance is suggested to be linked to cognitive decline [7] and Alzheimer’s disease via increased cerebrovascular inflammation [8,9]. In rodents, consumption of high-fat diet (HFD) and obesity have been shown to activate cytokine and inflammatory pathways in the hypothalamus—a brain region that plays a role in the regulation of glucose metabolism [10,11]. 

Preclinical studies show that the activation of the inflammatory processes in the hypothalamus is influenced by hormones released by adipose tissue and intestine and is initiated rapidly within a few days after high-fat feeding, already before the inflammatory events can be detected in peripheral tissues, such as in the liver [12,13]. Inflammation in the hypothalamus impairs a complex network of different cell types, the blood–brain barrier function, and the transfer of chemical signals from the cerebrospinal fluid to the central nervous system [14]. A recent study suggests that long-term HFD consumption leads to a metabolically activated state in brain-resident immune cells by cytokine signaling via interleukin-1β [15].

Excess consumption of saturated fat has also been linked with increased bacterial production of pro-inflammatory lipopolysaccharides in the gut, enhancing systemic inflammation [16]. Whether the regulation of whole-body glucose homeostasis initiates metabolic and inflammatory changes in gut microbiota, following metabolic signaling via the gut-brain axis and ending up with the interaction between brain and peripheral tissue, is unclear. It is also unclear how this crosstalk fails when the normal glucose homeostasis progresses to T2D. 

Regular exercise training is an effective way to improve skeletal muscle insulin sensitivity and reduce systemic inflammation [17,18]. Currently, it is poorly understood how exercise alters metabolism in organs and tissues that do not take part in skeletal muscle contraction, such as the brain and internal organs. Observed effects of exercise on internal organs include decreased liver and pancreatic ectopic fat content, improved beta cell function, modulated gut microbiota profile, and improved endotoxemia [19,20,21,22,23]. In addition, current evidence suggests that exercise training induces several neurobiological responses in the brain, enhances cognitive function and memory, and prevents aging-related degeneration and Alzheimer’s disease [24]. However, it is unclear whether exercise training can prevent and/or reverse brain inflammation and metabolic disturbances caused by obesity and insulin resistance.

In this study protocol, the aim is to examine insulin-stimulated GU in conjunction with low-grade inflammation with or without exercise intervention in an HFD-induced rat model of obesity. The study protocol is designed to investigate the early changes in the pathogenesis of obesity and insulin resistance simultaneously in the peripheral tissues and brain, as well as the effect of exercise training on the aforementioned conditions. This study protocol utilizes an HFD-induced rat model to study the effects of obesity and exercise training on glucose metabolism in the brain and peripheral tissues using [^18^F]FDG-PET method during hyperinsulinemic clamp. In addition, inflammatory responses are investigated using [^11^C]PK11195-PET as an estimate for inflammation in the brain. Until recent years, most of the preclinical PET data have been analyzed using semi-quantitative methods due to the needed high blood volume and laboriousness of attaining the arterial input function needed for full kinetic analysis. This CROSRAT study protocol uses state-of-the-art technology to measure blood radioactivity, enabling dynamic analysis of the small animal [^18^F]FDG-PET imaging data. This preclinical study complements the exercise intervention study we are performing in human monozygotic twin pairs discordant for BMI and/or insulin resistance (CROSSYS; NCT03730610) [25]. This preclinical design allows the study of tissue samples that cannot be obtained in humans and thus provides information on metabolic changes at the cellular and molecular levels. 

## 2. Experimental Design

The study groups and the overall experimental study design are presented in Figure 1. The rats are randomly divided into nine experimental groups that undergo (1) no intervention (^0^Chow), (2) 12-week intervention (^12^Chow, ^12^HFD, and ^12^(HFD+E)), (3) or a 24-week intervention (^24^Chow, ^24^HFD, ^12^HFD+^12^(HFD+E), ^12^HFD+^12^Chow, and ^12^HFD+^12^(Chow+E)). ^0^Chow is a baseline group with outcome measurements conducted at the age of 8 weeks. 

*12-week intervention*. During the first 12-week intervention, rats are on a high-fat diet in order to induce obesity and insulin resistance (^12^HFD) or on a simultaneous HFD and exercise training (^12^(HFD+E)) intervention to study whether training can prevent the negative effects of HFD. ^12^Chow serves as a control group. The groups are: age control group fed with normal chow for 12 weeks (^12^Chow), a diet intervention group fed with HFD for 12 weeks (^12^HFD), and an exercise training group fed with HFD for 12 weeks (^12^(HFD+E)). 

*24-week intervention.* The 24-week intervention groups are used to study whether the HFD-induced impairments in tissue metabolism can be ameliorated by exercise training and/or switch to a normal diet. The intervention groups are as follows: an age control group fed with normal chow for 24 weeks (^24^Chow), a diet intervention group fed with HFD for 24 weeks (^24^HFD), a diet switch group fed with HFD for 12 weeks, and then with normal chow for 12 weeks (^12^HFD+^12^Chow), an exercise training group fed with HFD for 24 weeks and exercised for the last 12 weeks (^12^HFD+^12^(HFD+E)), and an exercise training group fed with HFD for 12 weeks and then exercised and fed with normal chow for 12 weeks (^12^HFD+^12^(Chow+E)).

## 3. Materials

*Animals*. All animal experiments are approved by the State Provincial Office of Southern Finland (permission number ESAVI/4080/2019). Seven-week-old male Sprague Dawley rats (*n* = 144) are obtained from the Central Animal Laboratory of the University of Turku (Turku, Finland). The rats are housed in an environmentally controlled facility (12/12 h light-dark cycle, 21 °C, 55% humidity) in standard cages in groups of two to four rats with food and water provided ad libitum. Rats are randomly divided into experimental groups at the age of eight weeks.

## 4. Detailed Procedures and the Used Equipment

### 4.1. Diets and Voluntary Wheel Running

*Diets*. The rats are fed ad libitum without limitations either with standard low-fat chow (control diet, RM3 (E) Soya free, Special Diets Services, Shropshire, UK; 15.43 MJ/kg: 11.5% from fat, 27.0% from protein, 61.5% from carbohydrates (5.8% of which sugar)) or HFD (Western diet, 1.5% cholesterol, ssniff Spezialdiäten GmbH, Soest, Germany; 21.8 MJ/kg: 42.0% from fat (mainly composed of saturated fatty acids that originate from butter fat), 15.0% from protein, 43.0% from carbohydrates (34.1% of which sugar)) for 12 or 24 weeks.

*Voluntary wheel running*. The exercised rats undergo a 12-week voluntary wheel-running intervention. During the intervention, the rats are housed individually with access to running wheels (Intellibio, Seichamps, France) from 4 pm until 8 am (16 h a day) for four consecutive days a week, followed by three days of rest. The rest of the time, the rats are housed in groups in their home cages. The running time and distance are recorded daily with the activity wheel system and ActiviWheel software (v. 4.4., Intellibio, Seichamps, France).

### 4.2. Body Weight and Body Composition Analysis

The animals are weighed, and the body composition of the animals is analyzed with EchoMRI™ 700 Analyzer (EchoMRI LLC, Houston, TX, USA) at 0, 12, and 24 weeks of the intervention to measure total body fat and lean mass.

### 4.3. Fasting Blood Samples

Blood glucose is measured after a 4 h fast from the lateral tail vein with a glucometer (Contour XT, Bayer, Leverkusen, Germany) at 0, 12, and 24 weeks of intervention. Blood samples are collected into lithium heparin plasma collection tubes with PST™ plasma separator gel and serum collection tubes with serum separator SST gel (Microtainer^®^, BD, Franklin Lakes, NJ, USA). The samples are centrifuged for 90 s at 12,000× *g* (Eppendorf MiniSpin, Enfield, CT, USA) to separate the plasma and then stored at −80 °C for insulin and free fatty acid level assays. 

### 4.4. Fecal Samples 

Fecal samples are collected into Eppendorf^®^ Flex-Tubes^®^ at 0, 12, and 24 weeks and stored at −80 °C for microbiome and metabolomic studies.

### 4.5. Radiochemistry

[^18^F]FDG and [^11^C]PK11195 are synthesized at the Radiopharmaceutical Chemistry Laboratory at the Turku PET Centre. [^18^F]FDG is produced from mannosyl triflate with the FASTlab synthesizer (GE Healthcare, Waukesha, WI, USA). The radionuclide labeling is conducted via the nucleophilic substitution reaction as described previously with slight modifications [26]. [^11^C]PK11195 is synthesized according to the original synthesis method [27].

### 4.6. PET Imaging

All in vivo PET/computed tomography (CT) scans are conducted with the Inveon Multimodality PET/CT device (Siemens Medical Solutions, Knoxville, TN, USA), which has a spatial resolution of 1.3 mm. The scanner has an axial 12.7 cm field of view and 10 cm transaxial field of view, generating images from 159 transaxial slices. All cross-sectional in vivo PET/CT scans are conducted in isoflurane/oxygen gas mixture (2.5–4.0% in 700 mL/min oxygen flow) anesthesia on a temperature-controlled heating pad.

Arterial blood coincidences are recorded with Swisstrace Twilite II coincidence detector (Figure 2) and PMOD v. 4.1 (Psample, PMOD, Zurich, Switzerland), which enables the measurement of the whole blood arterial input function with a temporal resolution of one second without blood loss during the PET scan. Cross-calibration between Swisstrace Twilite II and Inveon Multimodality PET/CT scanner is performed prior to every experiment with the same tracer batch used for the injection. The arterial whole blood coincidences are corrected for the background signal counts, the time of the injection, radioactive decay, and the premeasured calibration factor between the coincidence detector and the Inveon Multimodality PET/CT scanner.

### 4.7. [^18^F]FDG-PET Study Combined with Hyperinsulinemic Euglycemic Clamp

At the end of the intervention, half of the rats (*n* = 8) from each of the nine groups underwent PET imaging with [^18^F]FDG combined with a hyperinsulinemic-euglycemic clamp to measure tissue-specific insulin-stimulated GU. The rat is weighed and administered 0.05 mg/kg buprenorphine for analgesia, after which the rat is anesthetized with an isoflurane/oxygen mixture (4% isoflurane for induction, 2.5% for maintenance). Surgical femoral artery and vein catheterization are performed for a 4 h fasted, anesthetized rat. The animal is transported to the scanner, and after a CT scan, the catheters are combined with a peristaltic pump and a Swisstrace Twilite II coincidence detector (Figure 2). Insulin clamp is initiated with a 120 mU/kg/min infusion for 3 min, followed by 60 mU/kg/min infusion until the blood glucose level drops to 6.0 mmol/L. Then, the insulin infusion is lowered to 18 mU/kg/min for the rest of the clamp protocol. Glucose infusion (20% volume/volume) is started at the same time. Blood glucose is measured (Contour XT, Bayer, Leverkusen, Germany) before the initiation of the clamp, every 3 min for the first 25 min and every 5 min onwards. After a steady blood glucose level of approximately 5 mmol/L is reached, a 45 min PET scan, with 30 × 10 s, 15 × 60 s, and 5 × 300 s framing, is started simultaneously with an intravenous (i.v.) injection of [^18^F]FDG (20 MBq). M-value, which depicts whole-body insulin sensitivity, is calculated as previously described [29].

### 4.8. [^11^C]PK11195-PET Study

At the end of the intervention, half of the rats (*n* = 8) from each of the nine groups undergo PET imaging with ^11^C-labelled R isomer of [1-(2-chlorophenyl)-N-methyl-N-(1-methylpropyl)-3-isoquinolinecarboxamide] ([^11^C]-(R)-PK11195) to study inflammation in brain.

Before the PET imaging, the rat is weighed and anesthetized with an isoflurane/oxygen mixture (4% isoflurane for induction, 2.5% for maintenance). After that, the lateral tail vein is cannulated. A 10-min CT scan is conducted for anatomical reference images and the attenuation correction of the PET imaging data. Then, [^11^C]-(R)-PK11195 (50 MBq) is administered intravenously, and a 30-min PET scan is started in tandem with 30 × 10 s, 15 × 60 s, and 2 × 300 s framing. 

### 4.9. Tracer Biodistribution Measurement and Ex Vivo Brain Autoradiography

*Tracer biodistribution*. After the [^11^C]PK11195 and [^18^F]FDG-PET scan, the animal is sacrificed with a cardiac puncture under terminal anesthesia. Blood is collected, and transcardial perfusion is performed; peripheral tissues and the brain are dissected, weighed, and measured for their radioactivity with an automatic γ-counter (WIZARD2, Perkin Elmer, Turku, Finland). The data are corrected for decay, and the time of injection and standardized uptake value (SUV) is calculated.

*Ex vivo autoradiography*. The brains of the animals scanned with [^11^C]PK11195 and [^18^F]FDG PET are snap-frozen in isopentane chilled with dry ice, after which 20 μm thick frozen coronal cryosections are cut with a cryostat microtome (Leica 3050S, Leica Biosystems, Richmond, IL, USA) on Superfrost^®^ Plus microscope slides (Thermo Fischer Scientific, Berlin, Germany). Digital autoradiography enables validating the PET imaging data by measuring the regional GU and TSPO ligand uptake at a higher spatial resolution of 25 µm.

Frozen tissue sections are cut, air-dried, and then exposed to imaging plates (Fuji Imaging Plate BASTR2025, Fuji Photo Film Co., Tokyo, Japan) for at least two half-lives of ^18^F or ^11^C depending on the tracer used. The radioactivity distribution on the plate is digitized using the BAS5000 analyzer (Fujifilm Lifesciences, Tokyo, Japan). Digital autoradiography data are analyzed using Aida Image Analysis software (Image Analyzer v. 4.22; Raytest Isotopenmeßgeräte GmbH, Straubenhardt, Germany) for count densities and expressed as background-erased photostimulated luminescence intensity per square millimeter (PSL/mm^2^) ratios relative to the selected reference region.

### 4.10. Tissue/Organ Samples

Harvested tissue samples and blood collected at the end of the in vivo PET/CT scans are further flash-frozen in liquid nitrogen and stored at −80 °C. 

### 4.11. PET Data Analysis

All PET imaging data are converted from 3D list mode to 2D sinograms by a Fourier rebinning algorithm and then reconstructed with a 2D-filtered back-projection algorithm into an image with a voxel size of 0.78 × 0.78 × 0.80 mm, or approximately 0.5 mm^3^.

For the brain analysis, the PET/CT images are pre-processed in MATLAB R2017a (The MathWorks, Natick, MA, USA) with an in-house semi-automated pipeline for preclinical images using SPM12 (Wellcome Department of Cognitive Neurology, London, UK) pre-processing functionalities and analysis routines. Heads are first cropped from the images, and individual PET images are co-registered via a rigid-body transformation with their corresponding CT scan. Next, the subject CT scans are co-registered with the T_2_-weighted W. Schiffer rat brain MRI template and volume of interest (VOI) atlas [30]. The combination of transformations is applied to the voxel size of 0.2  ×  0.2  ×  0.2 mm re-sampled PET images, matching the anatomical atlas dimensions. 

Tracer uptake in peripheral tissues and internal organs is analyzed using Carimas software (v. 2.10.3.0) that has been produced in-house. To calculate insulin-stimulated GU, fractional uptake ratio (FUR) is calculated as a ratio of average tissue radioactivity concentration post-injection and integral of blood TAC (recorded using Swisstrace Twilite II coincidence). FUR approximates the net influx rate (Ki) [31]. The insulin-stimulated GU is calculated using the following formula where Gluc = mean plasma glucose during the PET-scan, LC = Lumped constant, Density = density of the tissue.
GU = (Ki × Gluc)/(LC × Density)

For the [^11^C]PK11195 data, standardized uptake values (SUV) are calculated from the last 15 min of the scan with a following formula where C = mean radioactivity of the region of interest, dose = radioactivity of the injected tracer, weight = weight of the animal.
SUV = C/(dose/weight)

Subsequently, the [^11^C]PK11195-SUVs are normalized to the SUV of the whole blood obtained by cardiac heart puncture and measured using an automatic γ-counter (WIZARD2, Perkin Elmer, Turku, Finland) (decay correction to the injection timepoint) after the PET scan.

### 4.12. Brain Immunohistochemistry

The [^11^C]PK11195 uptake in the brain is validated with immunohistochemical stainings targeting reactive microglia and astroglia cells to confirm the neuroinflammatory profile of the animals. The localization of the reactive glial cells is visualized by staining the brain sections collected from ex vivo autoradiography. Fresh-frozen brain sections are air-dried, postfixed with 4% paraformaldehyde, and stained with antibodies against glial fibrillary acidic protein (GFAP, Agilent Dako, Z0334, Santa Clara, CA, USA) expressed in activated astrocytes and ionized calcium-binding adapter molecule 1 (anti-Iba-1, Wako 019-19741) expressed in reactive microglia.

Briefly, after the postfixation, the brain slides are incubated in 0.3% Triton X-100-based PBS blocking solution (2% BSA + 1.5% goat serum) for 30 min. The primary antibody incubation is carried out overnight with 1:500 dilution at +4 °C. After a wash with 1 × phosphate-buffered saline + 0.3% Triton X-100, the slides are incubated in the dark for 1 h with the secondary antibody (Alexa Fluor™ 568, Invitrogen, Waltham, MA, USA, dilution of 1:1000). The slides are further washed with 1 × phosphate-buffered saline + 0.3% Triton X-100, air-dried, and embedded in mounting medium (Vectashield HardSet, Newark, CA, USA). The sections are imaged with a 3DHISTECH MIDI Slide Scanner.

### 4.13. Statistical Analyses

Descriptive statistics of the cross-sectional data are expressed as mean (SD) for normally distributed data or as median (interquartile range, IQR) for non-normally distributed data. A normal distribution is examined using normality tests (Shapiro–Wilk) as well as visually from the residuals utilizing histograms and Q-Q-plots. Logarithmic or square root transformations are applied to gain normality of the data if applicable.

Longitudinal data, i.e., data that includes repeated measurements, such as body weight and composition, are examined using a mixed linear model, including one within-factor (time), one between-factor (group), and an interaction term (time × group) with the Kenward Roger correction (for degrees of freedom). In addition, contrasts are created between the groups at each timepoint. Data are reported as model-based means and 95% confidence intervals.

Differences between more than two groups in continuous variables in cross-sectional in vivo and ex vivo data are examined either with parametric (one-way ANOVA) or non-parametric (Kruskal–Wallis test). Appropriate post hoc analyses for individual parametric (Tukey’s or Dunnett’s comparison) and non-parametric (Dunn’s comparison) variance analyses are computed to examine paired differences within the data.

Experimental animals with missing data points are included in the statistical analysis using the restricted maximum likelihood estimation within the linear mixed models. Correlation analyses are performed using Pearson’s product-moment correlation coefficient for normally distributed data and Spearman’s rank correlation coefficient for non-normally distributed data. All statistical tests are performed as two-sided with the statistical significance level set at 0.05 using the SAS System, version 9.4 for Windows (SAS Institute).

## 5. Expected Results

This study enables the investigation of glucose metabolism and low-grade inflammation simultaneously in the brain and peripheral tissues in vivo using state-of-the-art imaging technology. In addition, this study provides ex vivo tissue samples for exploring the underlying mechanisms on a molecular level. This study will reveal novel information on pathophysiology related to obesity and by which mechanisms exercise may prevent or reverse these pathophysiological changes.

We hypothesize that HFD-induced obesity is associated with impaired whole-body and tissue level insulin sensitivity and worse blood glucose profile. Furthermore, it is expected that HFD increases insulin-stimulated GU in the brain, and the brain GU behaves opposite to GU in the skeletal muscle. Moreover, we hypothesize that HFD-induced increase in the brain GU is accompanied by increased brain inflammation that is measured comprehensively with (1) [^11^C]PK11195-PET, (2) autoradiography, and (3) immunohistochemistry. 

Importantly, we elucidate whether (1) exercise independently, (2) diet switching from HFD to normal chow, or (3) a combination of diet switch and exercise reverses the obesity-induced impairments in tissue-specific glucose metabolism and inflammation. We expect that all of these interventions will be effective in preventing the HFD-induced decrease in whole-body insulin sensitivity and skeletal muscle GU, as well as the increase in brain GU. Furthermore, we expect that these interventions also prevent HFD-induced inflammation in the brain. We hypothesize that the combination of diet switch and exercise is the most effective in preventing HFD-induced impairments, while exercise may have positive effects even without a change in the body composition of the rats.

According to the previous research conducted with spontaneously hypertensive normal-weight rats, 12 weeks of moderate-intensity involuntary treadmill running reduces neuroinflammation and microglial activation in the brain, as well as inflammation in the gut [32]. Data collected in vivo to support these findings are still scarce and sought after. A PET study using [^18^F]FDG and static image acquisition to determine the effects of regular aerobic exercise on cerebral glucose metabolism in different parts of the brain was conducted on healthy young-adult fasted rats after six weeks of involuntary treadmill running, revealing increased metabolism in brain regions associated with auditory processing, memory, motor function, and motivated behavior [33]. These findings showcase the effect that exercise has on the brain and underline the need for further research with more sophisticated methods to discover the underlying mechanisms. As of today, preclinical in vivo imaging data of inflammation related to obesity and exercise intervention does not exist. 

The strength of this study design is that it allows for the examination of the effects of both diet and exercise on glucose metabolism and low-grade inflammation in the context of obesity with two different timelines and endpoints. This provides insight into the timeline of the development of pathophysiological changes related to insulin resistance and inflammation and in which tissues these alterations occur first. The animal model of diet-induced obesity mimics the typical development of the condition, and related comorbidities manifested in humans better, as opposed to genetically modified models, like the Zucker Diabetic Fatty rat [34] or chemically induced models, where pharmaceuticals are used to cause pathological changes. Thus, the mechanisms underlying the pathophysiology of obesity-related comorbidities might be more similar to those in humans in this model. Animal models of obesity are crucial when studying the detailed mechanisms underlying the effects of weight gain and fat accumulation, as inducing obesity is ethically questionable in clinical settings, and many tissue samples are unattainable from humans.

The voluntary wheel-running exercise that was chosen in this study induces less stress compared with forced exercise [35], which contributes to the overall well-being of the laboratory animals and diminishes the effects of chronic stress on the study results. Forced exercise also poses the risk of over-training syndrome, having adverse effects on metabolism and overall health [35]. Although the animals need to be caged individually during the exercise session in order to record individual data, allowing the animals to socialize in between the exercise sessions reduces stress caused by single cage housing [36]. 

The animal model chosen for this study has its limitations, as the outbred rat strain (Sprague Dawley) allows for individual effects in response to the intervention [37]. Another limitation is the cross-sectional study design because the same animal cannot be imaged at multiple timepoints. The cross-sectional study design, however, provides tissue samples at multiple timepoints, which is not attainable without sacrificing the animal. In addition, while rodent models may broadly mimic humans, there are several key physiological differences to be taken into account when translating the results into the clinical setting. For example, even short-term fat-feeding protocols in rodents can produce marked hepatic lipid accumulation and hepatic insulin resistance without noticeable skeletal muscle insulin resistance [38,39], while skeletal muscle insulin resistance is thought to precede hepatic insulin resistance in humans [40,41]. However, even with limitations, the rodent models offer crucial data and tissue samples unattainable from humans, such as brain tissue.

CROSRAT is designed to shed new light on the underlying mechanisms of obesity-induced insulin resistance across the peripheral tissues and the brain. Furthermore, together with the complementary CROSSYS clinical intervention study, CROSRAT will offer new information on the mechanisms by which exercise prevents the development of obesity-induced impairments in insulin sensitivity in various tissues.

## Figures and Tables

**Figure 1 jfmk-09-00058-f001:**
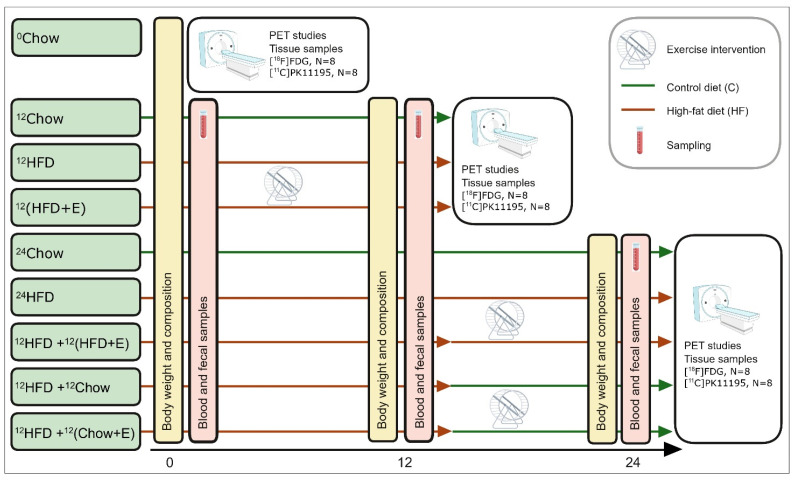
Experimental study design with the timeline in weeks, collected samples, and main procedures. Fecal samples are collected at the same time point as blood samples. ^0^Chow, a baseline group fed with normal chow; ^12^Chow, an age control group fed with normal chow for 12 weeks; ^12^HFD, a diet intervention group fed with high-fat diet (HFD) for 12 weeks; ^12^(HFD+E), an exercise training group with a 12-week exercise and HFD in parallel; ^24^Chow, an age control group fed with normal chow for 24 weeks, ^24^HFD, a diet intervention group fed with HFD for 24 weeks; ^12^HFD+^12^(HFD+E), an exercise training group fed with HFD for 24 weeks and exercise for the last 12 weeks; ^12^HFD+^12^Chow, diet switch group fed with HFD for 12 weeks and then normal chow for 12 weeks; ^12^HFD+^12^(Chow+E), an exercise training group fed with HFD for 12 weeks and exercise and normal chow for 12 weeks.

**Figure 2 jfmk-09-00058-f002:**
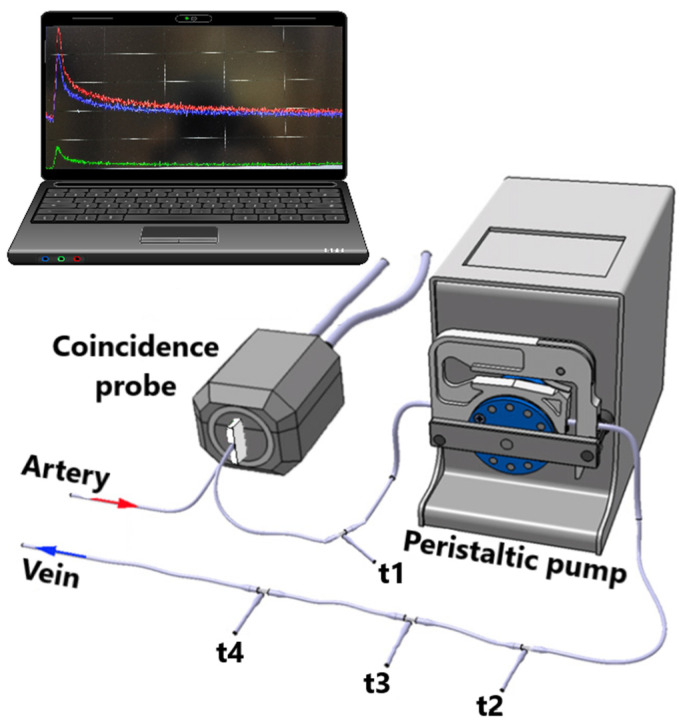
The experimental shunt set up used with positron emission tomography. The t-valves t1, t2, t3, and t4 are used for manual glucose level blood sampling, insulin infusion, glucose infusion, and tracer administration, respectively. Adapted from Ref. [28].

## Data Availability

The dataset generated and analyzed is available once the whole dataset is collected from the corresponding author at a reasonable request for researchers who have institutional review board/ethics approval and an institutionally approved study plan when applicable.

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
