# Peer review of "Mechanisms Leading to Increased Insulin-Stimulated Cerebral Glucose Uptake in Obesity and Insulin Resistance: A High-Fat Diet and Exercise Training Intervention PET Study with Rats (CROSRAT)"

_jfmk, 2024, doi:10.3390/jfmk9020058_

Round 1

Reviewer 1 Report

Comments and Suggestions for Authors

1. Lines 51-73: break up into separate paragraphs (>2).

2. How were the time periods determined? Why is a 12-week, for example is considered optimal to examine short-term effects of exercise? Was this related to studies on humans?

3. How informative would the measurement of HOMA-IR, C-peptide, and HbA1c be to interrogate glucose metabolism in the brain?

4. Justification for rat age is needed. Why 8 weeks?

5. "The groups that belong to the long intervention are also 169 measured in the middle of the intervention at 12 weeks" - why in the middle and not at the end?

6. Why plasma and not serum? And why heparinized plasma in particular?

7. Why is normality tested visually and not by normality tests (e.g., Kolmogorov-Smirnov test, Shapiro-Wilk, Anderson-Darling test, D'Agostino-Pearson omnibus test... etc.)?

8. Please mention that ANOVA and KW are used when comparing more than two groups. For two groups, t-test or Mann-Whitney U test are appropriate.

Reviewer 2 Report

Comments and Suggestions for Authors

This paper shows interesting new results regarding associated mechanism in cerebral glicose uptake in a rap model of HFD. The protocol of the present study seems adequate to meet the demands of the project and all analyzes were correctly described.

Reviewer 3 Report

Comments and Suggestions for Authors I consider this protocol very relevant and promising on the topic: “Mechanisms that lead to increased brain glucose uptake stimulated by insulin in obesity and insulin resistance: a PET intervention study with a high-fat diet and physical training with rats (CROSRAT)” . The protocol of this study was designed to investigate the initial changes in the pathogenesis of obesity and insulin resistance simultaneously in peripheral tissues and the brain, and the effect of physical training on the aforementioned conditions. This preclinical design allows the study of tissue samples that cannot be obtained in humans and thus provides information about metabolic changes at the cellular and molecular level. In my opinion, it presents a very relevant methodology with high scientific rigor, selecting instruments considered cutting-edge and not always easily accessible.   In summary: I reinforce the importance of this protocol, which is expected to produce new information about the pathophysiology related to obesity and understand which various mechanisms and exercise can prevent or reverse these pathophysiological changes. And that will provide answers to the proposed hypotheses: is obesity induced by HD associated with impaired insulin sensitivity throughout the body and at tissue level and a worse blood glucose profile. Furthermore, HD is expected to increase insulin-stimulated GU in the brain, and brain GU is expected to behave oppositely to GU in skeletal muscle. Furthermore, they hypothesize that the HD-induced increase in brain GU is accompanied by increased brain inflammation. I highlight the fact that the authors mention the limitations given that the sample must be mice. I consider this entire methodology to be well organized and at a very relevant level, with great scientific rigor. I therefore make suggestions that I believe could be a small contribution to the Protocol: Item 57 ……..high-fat diet (DH)……what type of fat? Saturated fat or others, this is a very important aspect. The rats are fed ad libitum………but is there a limit? which will be the amount of food given to you!!! It was important to know the amount consumed and the type of fat.    
